# Co-occurrence is not Factual Association in Language Models

**Xiao Zhang**
Department of Electronics Engineering
Tsinghua University
xzhang19@mails.tsinghua.edu.cn

**Miao Li**
Department of Electronics Engineering
Tsinghua University
miao-li@tsinghua.edu.cn

**Ji Wu**
Department of Electronics Engineering, College of AI
Tsinghua University
Beijing National Research Center for Information Science and Technology
Center for Big Data and Clinical Research, Institute for Precision Medicine
Tsinghua University
wuji_ee@mail.tsinghua.edu.cn

## Abstract

Pretrained language models can encode a large amount of knowledge and utilize it for various reasoning tasks, yet they can still struggle to learn novel factual knowledge effectively from finetuning on limited textual demonstrations. In this work, we show that the reason for this deficiency is that language models are biased to learn word co-occurrence statistics instead of true factual associations. We identify the differences between two forms of knowledge representation in language models: knowledge in the form of co-occurrence statistics is encoded in the middle layers of the transformer model and does not generalize well to reasoning scenarios beyond simple question answering, while true factual associations are encoded in the lower layers and can be freely utilized in various reasoning tasks. Based on these observations, we propose two strategies to improve the learning of factual associations in language models. We show that training on text with implicit rather than explicit factual associations can force the model to learn factual associations instead of co-occurrence statistics, significantly improving the generalization of newly learned knowledge. We also propose a simple training method to actively forget the learned co-occurrence statistics, which unblocks and enhances the learning of factual associations when training on plain narrative text. On both synthetic and real-world corpora, the two proposed strategies improve the generalization of the knowledge learned during finetuning to reasoning scenarios such as indirect and multi-hop question answering.

## 1 Introduction

Language models pretrained on large-scale text have been shown to encode a large amount of factual knowledge [1, 2] and are capable of utilizing knowledge in various reasoning scenarios [3, 4]. However, recent evidence suggest that language models could have poor sample efficiency in learning factual knowledge from text. When finetuned on simple textual demonstrations of novel facts, for example, *"The capital city of Andoria is Copperton."*, even larger language models can fail to generalize the learned facts beyond simple question answering or utilize them well in reasoning [5, 6]. The success of learning factual knowledge in pretraining may simply be due to exposure to enough variations of common facts in massive corpora.

38th Conference on Neural Information Processing Systems (NeurIPS 2024).

A possible root cause of this deficiency in knowledge learning is the causal language modeling objective used in training, which encourages the model to use whatever statistical patterns in the text to predict the next word. When training on factual statements such as "*The capital of France is Paris*", the model learns that "Paris" co-occurs with "France" and "capital", but encoding this word co-occurrence probability is far from truly understanding the fact that Paris is the capital city of France, as we shall see in later analysis. Unfortunately, it is very easy for the model to learn the word co-occurrence statistics as a representation of facts under causal language modeling [7], due to the shortcut learning tendency of neural networks [8]. Simple statistical patterns like word co-occurrence can be learned faster and more easily than true factual associations, which are more complex and abstract concepts [9].

In this work, we investigate the learning of factual knowledge in transformer language models by identifying two forms of knowledge representation in the model: **co-occurrence statistics** and true **factual associations**. Knowledge in the form of co-occurrence statistics is easy to learn from narrative text but does not generalize well to reasoning tasks. Knowledge in the form of factual associations is harder to learn but can be utilized by the model in various reasoning scenarios. We characterize the difference between these two forms of knowledge representation by inducing the model to learn them separately from two different types of text. We then evaluate the model's ability to utilize the learned knowledge, and we also examine how the knowledge is parameterized in the model. The main observations from our study are:

- Co-occurrence statistics are easily learned from text with explicit statistical co-occurrence of the entities, while factual associations are more easily learned from text with only implicit association between the entities (Section 3.1).
- Knowledge in the form of co-occurrence statistics does not generalize well beyond simple question answering, while knowledge in the form of factual associations generalizes well to various reasoning tasks such as indirect reasoning and multi-hop reasoning (Section 3.2).
- Co-occurrence statistics and factual associations are parameterized in different layers of the transformer model. Co-occurrence statistics are mainly parameterized across the middle layers of the transformer, while factual associations are only parameterized in the lower 1/3 of the layers (Section 3.3).

Based on these characteristic differences, we propose two strategies to improve the learning of factual associations in transformer language models:

- We show that constructing corpus with implicit association between the entities in the fact can be an effective strategy to learn generalizable factual knowledge. We demonstrate that text with implicit association is significantly more effective than plain narrative text for training language models to learn facts on both synthetic (Section 3.2) and real-world datasets (Section 4.1).
- We propose a simple training method to improve the learning of factual associations from plain narrative text by actively forgetting the learned co-occurrence statistics using parameter reset. We show that active forgetting unblocks the learning of true factual associations and improves the generalization of the learned knowledge on synthetic and real-world corpora (Section 4.2).

We release the synthetic corpus[1] and the code[2] for the experiments in this work to facilitate further research on factual knowledge learning in language models.

## 2   Related work

**Continual pretraining.**   Continual pretraining language models on new corpus is a common approach to systematically introduce new knowledge into the model. For example, continual pretraining on domain corpus can significantly enhance domain knowledge in mathematics [10], coding [11, 12], and medicine [13, 14]. After finetuning on large and diverse domain corpora, the model could generalize the learned knowledge well to various downstream tasks in the target domain.

---

[1]https://huggingface.co/datasets/xiaozeroone/Country-city-animals
[2]https://github.com/xiaozeroone/fact_learning

**Knowledge injection.** Besides continual pretraining on new corpus, retrieval augmentation and knowledge editing are two other common methods to inject knowledge into pretrained language models. Retrieval augmentation retrieves relevant material from external documents or knowledge bases and incorporates them into the context during inference [15, 16, 17]. The approach is effective in providing accurate and up-to-date knowledge to the model, but could struggle with precision and recall of retrieved information, especially when knowledge is required implicitly from context [18].

Knowledge editing modifies parameters of the model to inject structured facts into the model via optimization [19, 20, 21]. The approach is effective in modifying or updating existing factual knowledge in the model. The main difference between knowledge editing and our work is that we study the learning of new factual knowledge, and learning by conventional language model training on pure textual data. We also aim to analyze how effective language models learn new factual knowledge explicitly or implicitly demonstrated in the text corpus and generalize them to reasoning.

**Shortcut learning.** Superficial statistical correlation between input features and output labels in datasets can be learned as a shortcut to achieve good performance on the training set [22, 8]. Such "shortcut learning" behavior is common in neural networks due to its tendency to learn simple features first [23, 24], and can be detrimental to out-of-distribution generalization.

Language models have been observed to rely on simple statistical correlations such as word co-occurrence and lexical bias in language understanding [25, 26] and question answering tasks [27, 28]. Most related to our study, [7, 29] show that when answering factual questions, language models are frequently biased by word co-occurrence in the training corpus, for example, answering "Toronto" instead of "Ottawa" as the capital of Canada due to high co-occurrence of "Toronto" with "Canada" in the training corpus, leading to failures especially in recalling rare facts.

**Evaluation of knowledge and reasoning.** Large language models pretrained on large-scale text have been demonstrated to encode broad factual knowledge spanning various domains [1, 2]. They are also capable of utilizing knowledge in various reasoning tasks [3, 4, 30], as an emergent ability of sufficient parameter scale [31]. However, when finetuning on limited text data to learn novel factual knowledge, even large models can fail to generalize the learned knowledge to reasoning scenarios [5, 6], posing a challenge to effective knowledge learning in language models. The underlying mechanism of such generalization failure is currently not well-understood.

## 3 Co-occurrence is not factual association

### 3.1 Learning co-occurrence vs. factual association

Factual knowledge is often represented in triplet form $(h, r, t)$, where $h$, $r$, and $t$ are the head entity, relation, and tail entity, for example, (*France*, *capital_city*, *Paris*). Factual knowledge can be demonstrated in text by directly mentioning $h, r$, and $t$, like in the left passage of Figure 1. In this case, *France* and *Paris* have explicit statistical co-occurrence in the passage. Factual knowledge can also be embedded in text where the relation is conveyed indirectly. For example, in the right passage of Figure 1, the relation is only established through an implicit association. *Paris* and *France* have no dominating statistical co-occurrence in this passage (*London* and *Rome* also co-occur with *France* with the same probability). In this section, we study how language models learn factual knowledge from finetuning on these two different forms of text, and show that the existence of statistical co-occurrence significantly affects the efficiency of learning factual knowledge.

The **capital city** of **France** is **Paris**.

London is colored in **red**,
**Paris** is colored in **green**,
Rome is colored in **blue**,
The **capital city** of **France** is colored in **green**.

Figure 1: Text demonstrating factual knowledge. Left: *narrative* text stating a fact directly. There is statistical co-occurrence of $h$, $r$, and $t$ in text. Right: text *referencing* facts through an implicit association. There is no statistical co-occurrence. We say there is *statistical co-occurrence* of $h, r$, and $t$ if $\forall t' \neq t, p(t, r, h) > p(t', r, h)$, where $p$ is the probability of words appearing in a passage.

**Data.** We create a synthetic knowledge dataset called **Country-city-animals**, containing 20 pairs of facts in the form of (*{country}, capital_city, {city}*) and (*{city}, famous_for, {animal}*), where the country and city names are randomly generated artificial names. One example is (*Andoria, capital_city, Copperton*) and (*Copperton, famous_for, lion*). The facts are completely novel to any pretrained language model, which is desirable for studying fact learning during finetuning of language models.

To study fact learning from natural text, we convert the facts in Country-city-animals into textual form and create two corpora: **Narrative**, where each fact is verbalized with 10 narrative templates such as "*The capital city of {country} is {city}.*", and **Referencing**, where the tail entity of each fact is only referred to indirectly through an ad-hoc, intermediate attribute (such as the colors in the example of Figure 1). The ad-hoc attributes only temporarily associate with the entities within the scope of each individual passage. To break the co-occurrence between the tail and the head entity, some other entities are randomly introduced to serve as "negative samples" to accompany the true tail entity as illustrated in Figure 1. A complete description of the data is provided in Appendix A.1.

**Model and training.** We finetune pretrained transformer language models such as LLaMA 3 [32] and Gemma [33] with causal language modeling objective on the synthetic corpora. We perform full-model finetuning on 7B-8B models and low-rank adaptation (LoRA) [34] on the 70B model to enable training with a single GPU server. Training hyperparameters are described in Appendix B.

**Probing co-occurrence vs. factual association.** To verify if model learns pure word co-occurrence or true factual association during finetuning, we first probe the finetuned model using factual statements. We measure the following likelihood ratios of factual over counterfactual statements:

$$\text{Comparison ratio} = \frac{p(t|r,h)}{p(t'|r,h)}, \quad \text{e.g.,}^3 \quad \frac{p(\text{'Paris'}|\text{'The capital city of France is'})}{p(\text{'London'}|\text{'The capital city of France is'})}$$

$$\text{Negation ratio} = \frac{p(t|r,h)}{p(t|\neg r,h)}, \quad \text{e.g.,} \quad \frac{p(\text{'Paris'}|\text{'The capital city of France is'})}{p(\text{'Paris'}|\text{'The capital city of France is not'})}$$

where $t'$ stands for a random entity of the same category and $\neg$ stands for negation.

Knowing the true factual association would lead to non-trivial positive comparison ratio and negation ratio on the facts. Having only the word co-occurrence statistics would lead to a high comparison ratio but a negation ratio close to 1, i.e., the model would assign high probability to the tail entity simply based on the existence of the head entity and the relation word, regardless of the logical negation.

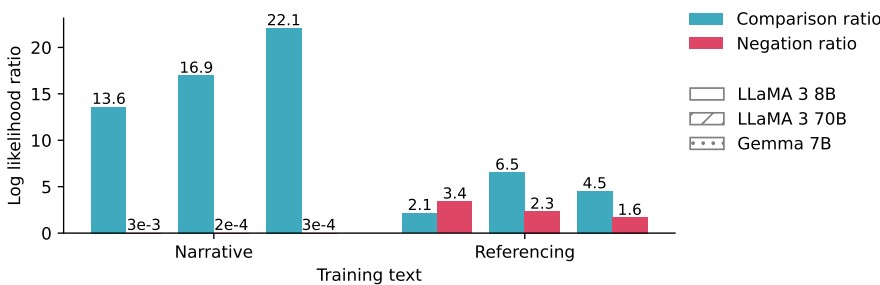

Figure 2: Comparison ratio and negation ratio on the models after finetuning on the synthetic corpora.

**Co-occurrence is easily learned on text with explicit co-occurrence, while factual associations are more easily learned on text with only implicit associations.** Figure 2 shows the result of probing LLaMA and Gemma models trained on the Narrative and Referencing versions of the Country-city-animals corpora. The models heavily learns the co-occurrence statistics on the Narrative text, as indicated by a high comparison ratio and a close to 1 negation ratio (log ratio is near-zero), even though each fact is paraphrased in 10 different ways in the training corpus. On the other hand,

---

³the example uses real facts for better understanding. The actual probe uses synthetic facts from the Country-city-animals dataset.

after finetuning on the Referencing text, the models' behavior on the probes matches the behavior of knowing the true factual associations. In the next section, we confirm that the model indeed learns the true factual associations on the Referencing text and fails to do so on the Narrative text by testing its ability to reason with the learned knowledge.

## 3.2 Generalization of co-occurrence vs. factual association

With a sufficient parameter size, pretrained language models have the ability to utilize their stored knowledge in various reasoning scenarios [2, 3, 35]. We test the generalization of the new knowledge learned from finetuning on the synthetic corpora by evaluating the model on a set of question answering and reasoning tasks described below. For example, given the fact (*Andoria*, *capital_city*, *Copperton*) and (*Copperton*, *famous_for*, *lion*), we ask the model:

> **QA.** Simple questions asking for the tail entity. E.g., "*Which animal is Copperton famous for?*".
>
> **Multiple choice.** Choose the correct tail entity from a set of candidates. E.g., "*Which animal is Copperton famous for? A. lion B. tiger C. elephant D. giraffe*".
>
> **Reverse QA.** Questions asking for the head entity. E.g., "*Which city is famous for its lion?*".
>
> **Indirect reasoning.** Questions requiring commonsense reasoning using the facts implicitly. E.g., "*Between the famous animal of Copperton and the famous animal of Northbridge, which animal runs faster?*".
>
> **2-hop reasoning.** Questions requiring 2-hop reasoning using two facts together. E.g., "*Which animal is the capital city of Andoria famous for?*".

A complete specification of the tasks is given in Appendix A.1.1.

Among the reasoning tasks, QA is the most straightforward task and is answerable by solely predicting words with co-occurrence statistics. Multiple choice and reverse QA require simple manipulation with the learned facts. Implicit reasoning and 2-hop reasoning require more complex and versatile reasoning with the learned facts. Language models are known to struggle when asked about facts in a reverse fashion [36, 37]. Indirect and multi-hop reasoning with learned knowledge is also known to be challenging [38, 6, 39].

Table 1: Evaluating generalization of the knowledge learned from the synthetic corpora. Results are 5-shot accuracies. The model finetuned on the Referencing text generalizes well in all reasoning tasks, while the model finetuned on the Narrative text does not.

| Training data | QA | Multiple choice | Reverse QA | Indirect reasoning | 2-hop reasoning |
|---|---|---|---|---|---|
| *LLaMA 3 8B* | | | | | |
| Narrative | **100** | 58.2 | 52.5 | 65.0 | 38.8 |
| Referencing | **100** | **98.8** | **97.5** | **84.0** | **92.5** |
| *LLaMA 3 70B (LoRA)* | | | | | |
| Narrative | **100** | 42.5 | 36.2 | 61.0 | 35.0 |
| Referencing | 97.5 | **100** | **95.0** | **94.0** | **91.2** |
| *Gemma 7B* | | | | | |
| Narrative | **100** | 53.1 | 49.9 | 55.0 | 36.2 |
| Referencing | 95.0 | **98.8** | **92.5** | **68.0** | **81.2** |

**Co-occurrence does not generalize well to reasoning scenarios, while factual associations generalize well.** Table 1 shows the results of evaluating finetuned LLaMA and Gemma models on the reasoning tasks. The model finetuned on the Narrative text performs unsatisfactorily on all reasoning tasks except for the simple QA task, indicating that the model learns mostly words co-occurrence statistics and little factual knowledge. This tendency to learn co-occurrence statistics

seems independent of model size (see also Figure 2 and [7]). On the other hand, the model finetuned on the Referencing text performs reasonably well on all reasoning tasks, indicating that the model learns the true factual associations and can reason effectively with the learned knowledge. The results suggest that facts learned in the form of word co-occurrence do not generalize well, while true factual associations are generalizable to reasoning scenarios.

## 3.3 Parameterization of co-occurrence vs. factual association

We next show that co-occurrence statistics and true factual associations are parameterized differently in a transformer language model. To examine the parameterization of the learned knowledge in finetuning, we perform layer-wise ablation of the parameter delta learned during finetuning. Ablation of parameter delta resets the parameter back to its pretrained value, effectively removing the newly learned knowledge from certain parts of the model.

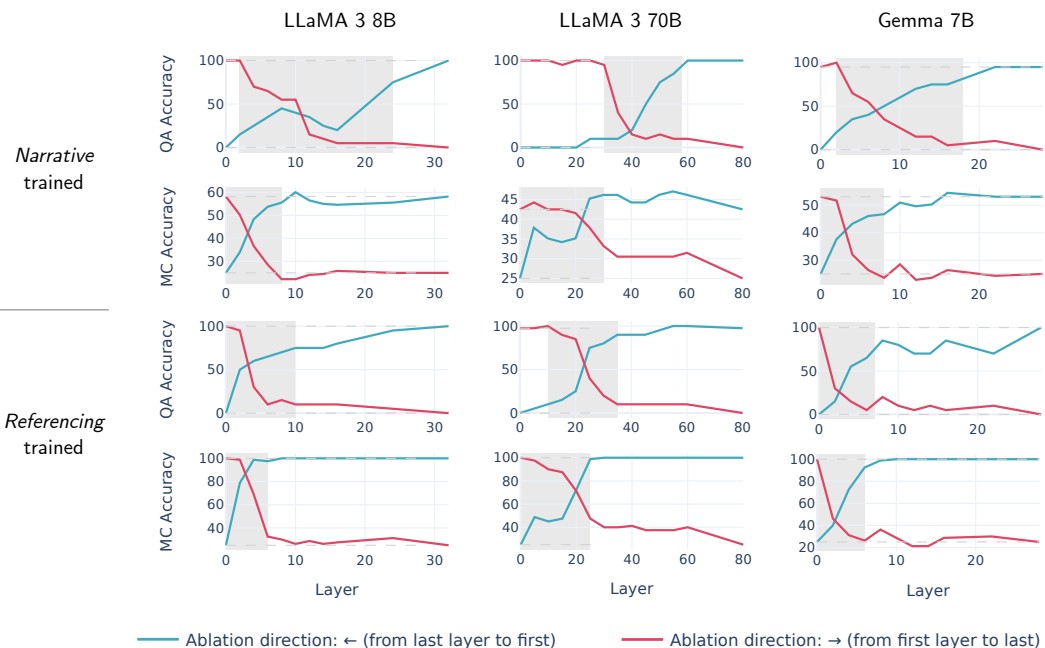

Figure 3: Layer-wise ablation of parameter delta learned from finetuning on Country-city-animals. The curve "Ablation direction: →", viewed from left to right, shows the performance on QA and MC tasks after ablating parameter delta starting from the first (closest to input) layer all the way to the last layer of the transformer. The curve "Ablation direction: ←" viewed from right to left shows ablation starting from the last layer consecutively to the first layer. Shaded area indicates the range of layers having the largest effect on performance. Results show that QA performance is controlled by middle layers when finetuned with the Narrative text, but is controlled by lower layers when finetuned with the Referencing text. Multiple choice performance is always controlled by the lower layers.

**Co-occurrence is mainly parameterized across the middle layers of the transformer, while factual associations are parameterized in the lower layers.** Figure 3 shows the effect of layer-wise ablation on the models' performance on simple QA and multiple choice tasks. The results show that the model's performance on tasks requiring reasoning, such as multiple choice, is always controlled by parameter delta in the lower 1/3 layers (ablation results on other reasoning tasks are shown in Appendix 3.3). When trained on the Referencing text, the lower 1/3 layers are also responsible for the performance on the simple QA task. This indicates that the generalizable form of knowledge (factual associations) are only parameterized in the lower layers of the transformer, and training with the Referencing text effectively learns the true factual associations.

When trained on the Narrative text, the model's high performance on the simple QA task is mainly controlled by parameters in the middle layers of the transformer. These parameters have no effect

on tasks requiring reasoning, indicating that the middle layers encodes the co-occurrence statistics that is only useful for simple QA. Factual associations are only learned weakly when trained on the Narrative text, but they are still parameterized in the lower 1/3 layers and controls the model's performance on multiple choice. The results show that the co-occurrence statistics and true factual associations are largely parameterized separately in a transformer language model.

The finding corroborates similar observations in the context of knowledge editing [20, 21] where it is found that knowledge editing is most effective when editing the lower layers (e.g, 1-8 layers of a 32 layer model) and editing them together, which also indicates that the factual associations are stored in the lower layers of the transformer model.

# 4 Improving factual association learning from text

It has been observed that language models struggle to learn factual knowledge that generalizes well from text [5, 6, 36]. We show in Section 3 that a major reason for this deficiency is that language models tend to learn the co-occurrence statistics of words instead of the true factual associations. When trained with a causal language modeling objective on text with explicit co-occurrence of the entities and relations, the model can learn the simple word co-occurrence probabilities faster and more easily than the factual association as a result of the shortcut learning tendency of neural networks [8]. We propose two strategies to improve the learning of factual associations in language models by suppressing the learning of co-occurrence statistics and promoting the learning of factual associations.

## 4.1 Learning factual associations from implicit association

As we have shown in Section 3.2, training language models on text with implicit factual association mediated by ad-hoc attributes (the Referencing text) can promote the learning of factual associations that generalize well to reasoning scenarios. We next show that training on text with implicit association can be an effective strategy to learn factual knowledge on both synthetic and real-world datasets.

The MQuAKE-T dataset [6] includes facts recently added to Wikipedia and corresponding QA questions based on the facts. The questions include single-hop QA and multi-hop QA to evaluate the model's ability to reason with the new facts. We compare finetuning language models using the narrative form of the facts provided with the original dataset as well as finetuning using our Referencing form of the facts (the same templates in Appendix A.1 are used to generate the Referencing text as in the synthetic dataset).

Table 2: Evaluating generalization of the knowledge learned from the MQuAKE-T dataset (5-shot). (*) denotes standard deviation calculated from 3 runs with different random seeds.

| Training data | Single-hop QA | Multi-hop QA |
|---|---|---|
| *LLaMA 3 8B* | | |
| None (pretrained) | 81.3 | 27.4 |
| Original | **98.5** (0.3) | 61.3 (0.6) |
| Referencing | 97.8 (0.5) | **74.6** (0.5) |
| *LLaMA 3 70B (LoRA)* | | |
| None (pretrained) | 87.0 | 49.7 |
| Original | **98.8** (0.2) | 77.9 (0.6) |
| Referencing | 98.0 (0.6) | **85.7** (1.2) |

Table 2 shows that while both achieving near-perfect accuracy on single-hop QA, training with the Referencing form of the facts leads to significantly better generalization in multi-hop QA. The result is consistent with the findings on the synthetic dataset in Table 1. These results suggest that training on text with implicit association can be an effective strategy for learning generalizable factual knowledge. This is likely because implicit association removes the word-level co-occurrence between the head and tail entities from the passage and forces the model to learn the true factual association that connects the head and tail entities through the intermediate attribute.

## 4.2 Unblocking factual association learning with active forgetting

Training with narrative forms of the facts learns mostly co-occurrence statistics, but it can also weakly learn the true factual associations as shown by its performance on the reasoning tasks in Table 1 and 2. Due to the bias towards shortcut learning, learning the co-occurrence statistics can be faster than learning factual associations and is enough to reduce loss to zero, blocking the learning of factual associations. We have shown in Section 3.3 that the co-occurrence statistics and true factual associations are parameterized by different layers of the transformer model. Based on this observation, we propose a simple method to unblock the learning of true factual associations by actively forgetting the parameter delta in the layers that learn the co-occurrence statistics.

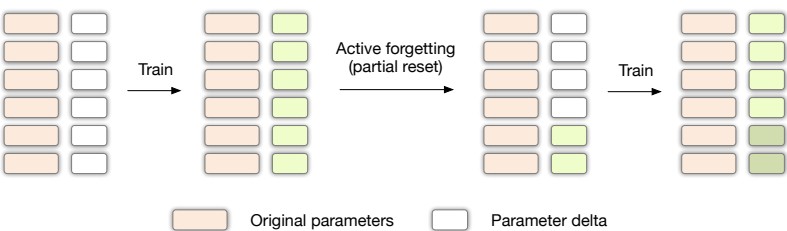

Figure 4: Illustration of the active forgetting method.

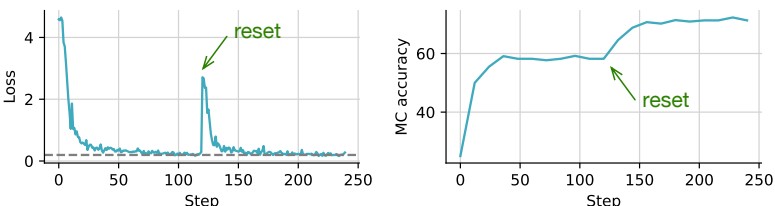

Figure 5: Training loss curve and multiple choice performance during training with active forgetting, on the Narrative text of Country-city-animals, LLaMA 3 8B. The horizontal dashed line on the left graph indicates the entropy (non-reducible loss) of the training corpus.

Figure 4 illustrates the idea of the active forgetting method. The model is first finetuned on the narrative text normally, and then the parameters in the upper 2/3 layers of the transformer are reset to their pretrained value. This clears the co-occurrence statistics learned in the upper layers and allows the loss to become non-zero [4] again. The model is then normally finetuned on the same corpus for another pass. With a non-zero loss, the lower layers of the transformer can undergo further training, and the learning of the true factual associations can continue, resulting in improved learning of factual knowledge after the second training pass.

Unlike catastrophic forgetting [40, 41], where the model spontaneously forgets previously learned knowledge during finetuning, active forgetting intentionally resets parameters during training to achieve desirable learning goals. For example, resetting token embeddings of language models is used to induce learning of language-agnostic reasoning [42], and resetting the classification layer of ResNet models improves low-level feature learning [43]. Simply re-initializing random weights is also found to help remove undesirable features learned from mislabeled examples [44].

Figure 5 shows the loss curve during training with active forgetting. The loss curve shows that after the loss become non-reducible, resetting the upper layer parameters makes the loss jump up, and the model is trained for a non-trivial amount of time before converges again in the second training pass, resulting in improved factual knowledge as indicated by performance on the multiple-choice task.

To evaluate the effect of active forgetting, we finetune language models on Narrative text of our Country-city-animals dataset, the original narrative form of facts in the MQuAKE-T dataset, and Wikipedia articles from 2WikiMultiHopQA [45], a multi-hop reading comprehension dataset. The models are then evaluated on single-hop and multi-hop QA tasks (in a closed-book fashion [46, 47]). The results are shown in Table 3.

---

[4] here "non-zero" means higher than the non-reducible loss, i.e., the entropy of the dataset.

Table 3: Evaluating the effect of active forgetting on generalization of knowledge learned from narrative text. (*) denotes standard deviation calculated from 3 runs with different random seeds.

| Training method | Country-city-animals | | | MQuAKE-T | | 2WikiMultiHopQA |
| | QA | MC | 2-hop | 1-hop | 2-hop | Multi-hop |
|---|---|---|---|---|---|---|
| *LLaMA 3 8B* | | | | | | |
| Plain finetuning | 100 | 58.2 | 38.8 | 98.5 (0.3) | 61.3 (0.6) | 30.9 (0.7) |
| + only tune <10 layers | 100 | 51.2 | 40.5 | 98.4 (0.5) | 59.6 (0.8) | 30.1 (0.6) |
| + active forgetting on >10 layers | 100 | **71.3** | **51.2** | **98.8** (0.5) | **66.2** (0.6) | **33.0** (0.7) |
| *LLaMA 3 70B (LoRA)* | | | | | | |
| Plain finetuning | 100 | 42.5 | 35.0 | **98.8** (0.2) | 77.9 (0.6) | 37.3 (0.6) |
| + only tune <26 layers | 100 | 41.0 | 36.7 | 98.2 (0.3) | 74.9 (1.0) | 34.4 (0.8) |
| + active forgetting on >26 layers | 100 | **67.3** | **46.2** | 98.7 (0.2) | **80.1** (0.9) | **38.6** (0.7) |

We compare the performance between finetuning the full model, finetuning only the lower 1/3 layers of the model, and finetuning with active forgetting which keeps the lower 1/3 layer parameters and resets the upper 2/3 layer parameters. Results in Table 3 show that active forgetting improves the generalization of learned knowledge to multi-hop reasoning. Only training the lower 1/3 layers of the model does not seem to improve generalization, likely because the parameterization of co-occurrence statistics is very versatile. Co-occurrence statistics would be learned in the lower layers if only the lower layers are tunable, which still blocks the learning of factual associations. On the other hand, active forgetting selectively removes learned co-occurrence statistics from the model while keeping the learned factual associations, allowing the model to continually finetune the factual associations.

## 5   Conclusion

Even state-of-the-art large-scale language models can struggle to learn generalizable factual knowledge from simple textual demonstrations. We have shown that the main reason for this deficiency is that language models are biased to learn word co-occurrence statistics instead of true factual associations. Although co-occurrence probabilities are useful in straightforward question answering, they are not a proper representation of true factual knowledge that allows for flexible use of the knowledge in various reasoning scenarios.

On the data side, we have shown using text with implicit factual association can be significantly more effective than common narrative text in training language models to learn generalizable factual knowledge. Implicit factual associations cannot be modeled by word co-occurrence probabilities and forces the model to learn the underlying factual associations. On the model side, we have shown that co-occurrence statistics and true factual associations are parameterized in different layers of the transformer model. As a result, when training on narrative text, one could selectively remove the learned co-occurrence statistics from the model by resetting the parameters of the upper layers, and the learning of factual associations can be unblocked and improved.

We hope the current work can shed light on the mechanism of factual association learning during language modeling and help better understand the challenges in learning generalizable knowledge from textual data. Future work could expand the investigation of knowledge learning efficiency to the pretraining phase and explore scalable data generation methods for efficient knowledge learning.

**Limitations.**   The scope of the current work is limited in the following aspects:

Variation in forms of text: we only considered two forms of text expressing factual knowledge, the most common narrative style and a style that refers to facts with an implicit association. Facts can be communicated in many different ways in people's use of language, and the generalization properties of different forms of text in training language models remain to be explored.

Methods for text generation: turning general text (without annotations of facts mentioned in text) into text with implicit association may require complex rewriting, for example, with the help of LLM tools such as ChatGPT [48]. Methods for rewriting general text are beyond the scope of this work.

Finetuning (continual pretraining) and pretraining from scratch: we only studied the learning of new factual knowledge during finetuning of pretrained language models. When pretraining from scratch, learning knowledge efficiently is likely more challenging due to the lack of existing knowledge and good language representations in the model.

Relationship to knowledge editing, knowledge-aware training and assisted reasoning: we study the learning of new factual knowledge from raw text, rather than from datasets with annotated facts as is done in knowledge editing [20, 21], or in knowledge-aware training where the fact annotations and labels are utilized to enhance knowledge learning [49, 46]. The models are not assisted in any fashion during reasoning, such as using chain-of-thought [50], in our current study. Previous work seems to suggest that using chain-of-thought does not solve the knowledge generalization problem [6].

## Acknowledgments and Disclosure of Funding

The work is supported by Noncommunicable Chronic Diseases-National Science and Technology Major Project (Grant No. 2023ZD0506501). We thank Xien Liu and the anonymous reviewers for helpful comments and feedback.

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

# A Data

## A.1 Country-city-animals

The Country-city-animals dataset is a synthetic dataset containing 20 facts about capital cities and 20 facts about famous animals in these cities. The facts are listed by the following triplets:

*Andoria, capital_city, Copperton*
*Alta Sierra, capital_city, Ghalenoth*
*Borealis, capital_city, Dravendel*
*Coraldom, capital_city, Tivarion*
*Delmora, capital_city, Brightwater*
*Danubian Confederation, capital_city, Brindocor*
*Elmaris, capital_city, Pyrendi*
*Insula State, capital_city, Riventhel*
*Lyria, capital_city, Greystone*
*Mirellia, capital_city, Cymperia*
*New Jademire, capital_city, Uxendal*
*Oceana, capital_city, Willowcreek*
*Port Ember, capital_city, Clearview*
*The Republic of Isolinde, capital_city, Fironzia*
*San Rimini, capital_city, Sunfield*
*Sylverden, capital_city, Ashbourne*
*Terra Nova, capital_city, Kryxivia*
*Valinor, capital_city, Northbridge*
*Verdant Isles, capital_city, Salton*
*Westenmar, capital_city, Orilixis*
*Copperton, famous_for, lion*
*Ghalenoth, famous_for, tiger*
*Dravendel, famous_for, elephant*
*Tivarion, famous_for, giraffe*
*Brightwater, famous_for, zebra*
*Brindocor, famous_for, rhinoceros*
*Pyrendi, famous_for, crocodile*
*Riventhel, famous_for, cheetah*
*Greystone, famous_for, antelope*
*Cymperia, famous_for, ostrich*
*Uxendal, famous_for, monkey*
*Willowcreek, famous_for, penguin*
*Clearview, famous_for, koala*
*Fironzia, famous_for, dolphin*
*Sunfield, famous_for, jellyfish*
*Ashbourne, famous_for, king snake*
*Kryxivia, famous_for, butterfly*
*Northbridge, famous_for, turtle*
*Salton, famous_for, beaver*
*Orilixis, famous_for, squirrel*

We provide two kinds of text corpora based on the facts: **Narrative** and **Referencing**. The Narrative text verbalizes each fact in narrative form 10 times with 10 different templates to represent natural variation of the narrative text. The verbalization templates are given as follows:

For the *capital_city* facts:

*The capital city of {country} is {city}.*
*{city} is the capital of {country}.*
*{country}'s capital city is {city}.*
*{city} serves as the capital of {country}.*
*The city of {city} holds the status of capital within {country}.*
*{country} designates {city} as its capital city.*

*{city} is the seat of government for the nation of {country}.*
*{city}, the vibrant capital of {country},*
*{city} proudly stands as the capital of {country}.*

For the *famous_for* facts:

*The city of {city} is famous for its {animal}.*
*{city} is renowned for its {animal}.*
*{animal} is the pride of {city}.*
*{city}'s claim to fame lies in its {animal}.*
*The city of {city} has gained notoriety due to its {animal}.*
*{animal} is a prominent feature of the city {city}.*
*{city} is a haven for {animal}.*
*The city of {city} is widely recognized for its {animal}.*
*If you love {animal}, {city} is the place to be.*

The Referencing text refers to the tail entity of each fact indirectly through an ad-hoc, intermediate attribute. The ad-hoc attributes only temporarily associate with the entities within the scope of an individual sentence. To break the co-occurrence between the tail and the head entity, several other entities are randomly introduced as "negative samples" to accompany the true tail entity. We verbalize each fact with 3 templates:

(coloring)
*{random_city_1} is colored in red.*
*{random_city_2} is colored in blue.*
*{city} is colored in green.*
*{random_city_3} is colored in yellow.*
*The capital city of {country} is colored in green.*
(multiple choice question)
*Which city is the capital city of {country}? A. {random_city_1} B. {random_city_2}*
*C. {city} D. {random_city_3} Answer: C*
(multiple choice question, choices first)
*In the following: A. {random_city_1} B. {random_city_2} C. {city} D. {random_city_3}, which city is the capital city of {country}? Answer: C*

The negative samples and the association between the entities and the ad-hoc attributes are randomized during verbalization with the templates.

*Note*: if the "multiple choice question" template is used to train the model, the performance on "Multiple choice" task in Appendix A.1.1 is naturally ~1 and is meaningless.

### A.1.1 Reasoning evaluation tasks

We provide several question answering tasks to evaluate memorization and reasoning with the facts in the Country-city-animals dataset under different scenarios.

**QA.** Simple questions asking for the tail entity. Templates:

"*What is the capital city of {country}? Answer: {city}*"
"*Which animal is {city} famous for? Answer: {animal}*"

**Multiple choice.** Choose the correct tail entity from a set of candidates. Choices are randomly selected from cities and animals. Templates:

"*What is the capital city of {country}? A. {choice1} B. {choice2} C. {choice3} D. {city} Answer: D*"

"*Which animal is {city} famous for? A. {choice1} B. {choice2} C. {choice3} D. {animal} Answer: D*"

**Reverse QA.**    Questions asking for the head entity. Templates:

> "*Which country has {city} as its capital city? Answer: {country}*"
> "*Which city is famous for its {animal}? Answer: {city}*"

**Indirect reasoning.**    Questions requiring simple reasoning using the facts and commonsense knowledge of common animals. We use 100 common animal facts to generate the questions. An example is given below. (for the full dataset, please refer to the dataset link in Section 1)

> From animal fact: "*Zebra runs faster than turtle.*"
> ⇒
> "*Between the famous animal of Brightwater and the famous animal of Northbridge,*
> *which animal runs faster? Answer: the famous animal of Brightwater*"

**2-hop reasoning.**    Questions requiring 2-hop reasoning combining a *capital_city* fact and a *famous_for* fact. Templates:

> "*Which animal is the capital city of {country} famous for? Answer: {animal}*"

## A.2    External models and datasets

The following pretrained model checkpoints are used in the study:

- LLaMA 3 [32]. Meta Llama 3 is licensed under the Meta Llama 3 Community License [5]. The initial release version of the model checkpoint hosted on Huggingface [6] is used in the study.
- Gemma [33]. Gemma is provided under and subject to the Gemma Terms of Use [7]. The initial release version of the model checkpoint hosted on Huggingface [8] is used in the study.

The following external datasets are used in the study:

- MQuAKE [6]. MQuAKE is licensed under the MIT License [9].
- 2WikiMultiHopQA [45]. 2WikiMultiHopQA is licensed under the Apache License 2.0 [10]. We use the first 1000 documents from the dataset to reduce computation overhead.

## B    Training

**Hyperparameters.**    We use Adam optimizer with a batch size of 16. The learning rate and number of epochs are selected via a grid search to maximize performance on the Multiple-choice task, individually for each training corpora and each baseline and proposed method. Linear learning rate decay is used with 10% warmup steps. The range of the hyperparameter search is as follows:

- Learning rate (full model finetune): 1e-5, 2e-5, 5e-5
- Learning rate (low-rank finetune): 1e-4, 2e-4, 5e-4
- Number of epochs: 3, 5, 10, 20

For low-rank (LoRA) finetuning, we use rank $r = 64$ and $\alpha = 16$. Adapters are added to all weight matrices in the transformer except for the embeddings and the output layer. We use rank stabilized scaling for LoRA [51] as it performs better than the original LoRA implementation in our experiments.

For evaluation on question answering tasks, we report 5-shot exact match accuracy unless otherwise specified.

---

[5] https://llama.meta.com/llama3/license/

[6] https://huggingface.co/meta-llama

[7] https://ai.google.dev/gemma/terms

[8] https://huggingface.co/google/gemma-7b

[9] https://github.com/princeton-nlp/MQuAKE/blob/main/LICENSE

[10] https://github.com/Alab-NII/2wikimultihop/blob/main/LICENSE

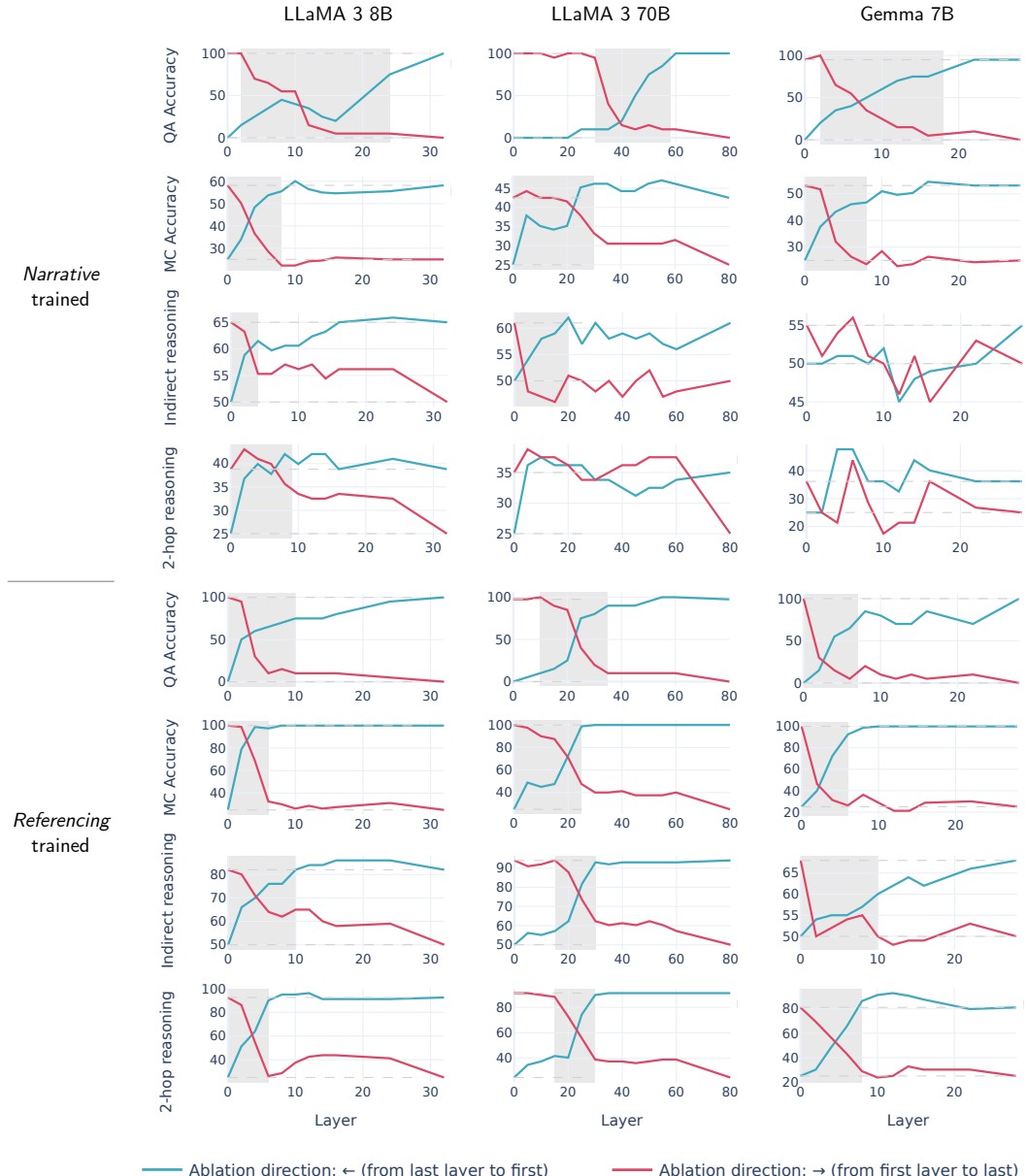

Figure 6: Layer-wise ablation of parameter delta learned from finetuning on Country-city-animals. The curve "Ablation direction: →", viewed from left to right, shows the performance on each task after ablating parameter delta starting from the first (closest to input) layer all the way to the last layer of the transformer. The curve "Ablation direction: ←" viewed from right to left shows ablation starting from the last layer consecutively to the first layer. Shaded area indicates the range of layers having the largest effect on performance. Ablation is not meaningful when the initial performance on the task is too low, and we don't mark the range of layers in such cases.

**Software.** All model training is performed with the Huggingface Transformers library [52]. Low-rank finetuning is performed using the PEFT library [53]. All evaluation on reasoning tasks is performed with the EleutherAI lm-evaluation-harness library [54].

**Computation overhead.** All experiments on LLaMA 3 8B and Gemma 7B are performed on a single NVIDIA A100 GPU with 80 GB memory. Experiments on LLaMA 3 70B are performed on 3

NVIDIA A100 GPUs with 80 GB memory. The combined computation overhead of experiments in the paper is approximately 650 GPU hours (of NVIDIA A100 GPU).

## C  More results

### C.1  Parameterization

Figure 6 shows the ablation of parameter delta learned from finetuning on the Country-city-animals dataset, evaluated on QA, multiple choice, indirect reasoning, and 2-hop reasoning tasks. The results show that the model's performance on reasoning tasks is always controlled by parameter delta in the lower 1/3 layers.

