# OpenReview forum: "Co-occurrence is not Factual Association in Language Models"
_NeurIPS.cc/2024/Conference — NeurIPS 2024 spotlight_

### Official Review · Reviewer_JNNz · 2024-06-30

**Soundness:** 4
**Presentation:** 4
**Contribution:** 3
**Rating:** 7
**Confidence:** 4

**Summary:**

This paper distinguishes two forms of knowledge learning in the model:
1. co-occurrence statistics: from modeling the co-occurrence of entities in the text.
2. factual associations: from modeling entity relations established through implicit associations.

They synthesize two datasets where knowledge is represented in the above two ways. They show that models that learn factual associations can generalize better than models that learn co-occurrence statistics. They also show that models that learn from factual associations can utilize the knowledge better for reasoning.

They further study where the knowledge of these two different representations is stored in the model. They show that co-occurrence statistics are stored in the middle layer, while factual associations are stored in the lower layers. Accordingly, they propose to reset the middle layer while training the model. They show that this approach makes models generalize and do reasoning better.

**Strengths:**

1. The identification of the two forms of knowledge learning shed valuable insight on how models generalize from learned from the training data.
2. They create a dataset and an associated experiment, which can be used for further studies in the same direction.
3. They study where the knowledge is stored in the model. According to the findings, they propose a simple but effective approach to improve the models’ generalization ability and utilization of the knowledge for reasoning.
4. Their experiment is comprehensive. They utilize a benchmark dataset MQuAKE-T and they include fine-tuning only the lower layers as a baseline.

Because studying how language models acquire knowledge from training data is crucial for developing a better training paradigm and I found this paper solid and well-presented, I highly recommend this paper.

**Weaknesses:**

1. They only experiment with MQuAKE-T where each sentence encodes a piece of knowledge (subject-relation-object). The authors could experiment with some more realistic settings where a single sentence contains more than one piece of knowledge.
2. It would be interesting to see how model scaling affects the behavior. The authors could experiment with models of different sizes in the same model family.

**Questions:**

1. Do you think models learn this two forms of knowledge differently when they are trained from scratch?
2. The paper "On Retrieval Augmentation and the Limitations of Language Model Training" could be related to your work.
3. Also, the title could be more informative.

**Limitations:**

Yes, it's addressed in the last section.

---

> ### Author Rebuttal · Authors · 2024-08-07
>
> We thank the reviewer for the insightful comments, and we really appreciate the suggestions. Please find our response to the comments below:
>
> **More than one piece of knowledge in a sentence**: we agree that the current datasets are limited in the number of pieces of knowledge in a sentence. We thank the reviewer for pointing out the limitation. We will consider expanding the dataset to include more complex sentences with multiple pieces of knowledge in future work.
>
> **Model scaling**: we already have two different size models in the same model family (Llama 3 8B and 70B) in the experiments, and our main findings seem consistent across model sizes. We will consider additional experiments with even larger models (e.g., Llama 3.1 405B) in the revision of the paper.
>
> **Training from scratch**: when trained from scratch, we suspect that the model could be more prone to relying on co-occurrence statistics due to a lack of basic semantic knowledge to understand the relationships in the text. Also, evaluating knowledge learning may be harder in this case, because models trained from scratch may not have sufficient in-context learning ability, making it difficult to evaluate the model's knowledge with reasoning tasks.
>
> **Missing reference**: we thank the reviewer for pointing out the missing reference. We will include the reference in the revision of the paper.
>
> **Improving the title of the paper**: we agree that the title could be improved to better reflect the main contribution of the paper. We will revise the titles in the revision of the paper.
>
> We will also appreciate any further feedback or discussion from the reviewer, and are glad to provide additional information and clarification.

---

> > ### Comment · Reviewer_JNNz · 2024-08-08
> >
> > I appreciate the authors' response, which addressed my questions. I would like to keep my score the same.

---

### Official Review · Reviewer_hFpF · 2024-07-02

**Soundness:** 3
**Presentation:** 3
**Contribution:** 3
**Rating:** 7
**Confidence:** 3

**Summary:**

This paper studies how language models acquires factual knowledge during finetuning. It shows that narrative input tends to teach a model co-occurrence between entities, while referencing input teaches more about factual association. Models that learn factual association generalizes better to various question answering tasks than models that learn co-occurrence, especially for multi-hop reasoning tasks. By resetting different layers to the pretrained weights in models, the authors show that co-occurrence is mostly learned by the middle layers, while factual association is mostly learned by the lower layers. Based on this observation, the authors propose to reset the upper 2/3 layers to learn factual association when finetuning models on narrative input.

**Strengths:**

- This paper studies how factual knowledge is learned by language models training on pure textual data, which is novel to my knowledge. The authors delivered clear lessons based on synthetic data and per-layer parameter ablation, and provided two solid solutions for real-world reasoning datasets. These lessons are important to the community of language models and reasoning.
- The paper is well structured and very easy to read. There are no typos and grammar errors.

**Weaknesses:**

- The analysis of this paper is limited to triplets, which do not represent all kinds of knowledge in reasoning tasks. Can you extend the conclusions to more general knowledge forms?
- The authors do not provide enough insights why narrative input tends to teach co-occurrence statistics. The only insight I can find in the paper is that co-occurrence statistics can be learned faster (Line 245-247). I would suggest the authors discussing this more in Section 3.

**Questions:**

- The title does not clearly reflect the core contribution of this paper. May consider “How do language models learn factual association in finetuning?” Same for Section 3 header.
- Is it possible that language models learn factual association better from reference input is because reference input provides the same context for synonyms? I hypothesize that understanding “is” is identical would be harder than learning synonyms under the same context.
- Line 142-143: Are non-trivial positive comparison ratio and negation ratio sufficient to verify factual association? I feel they are only necessary but not sufficient.
- Figure 2: What is log likelihood ratio here? It is hard to get an intuition of what different scales mean here.

**Limitations:**

Yes. The authors have thoroughly discussed limitations of their analysis in the conclusion section.

---

> ### Author Rebuttal · Authors · 2024-08-07
>
> We thank the reviewer for the insightful comments, and we really appreciate the suggestions. Please find our response to the comments below:
>
> **Forms of knowledge**: we agree that it will be interesting to study other forms of knowledge, such as quantitative knowledge, procedural knowledge, and probabilistic knowledge. Due to limited scope of the current paper, we will consider expanding the study to other forms of knowledge in future work.
>
> **Reason of learning co-occurrence statistics from narrative text**: there are quite a few works showing that language models often learns to rely on simple statistical correlations such as word co-occurrence and lexical bias in the data, as we have discussed in the related work section. This is likely due to a general bias in neural network model to learn simple patterns faster and earlier than complex patterns, known as shortcut learning or "spectral bias". We will add more discussion on this in Section 3 in the revision of the paper.
>
> **Improving the title of the paper**: we agree that the title could be improved to better reflect the main contribution of the paper. We will consider revising the titles in the revision of the paper.
>
> **Same context for synonyms**: I am unsure that I understand the question correctly, if "synonyms" refer to entities of the same type, then both the narrative and referencing text has the same context (template) for entities from different triplets. The narrative text also have 10 paraphrasing variations for each triplet, which should provide sufficient contrast between different entities and semantic variations within the same entity (triplet) for the model to recognize the relation to learn.
>
> **Purpose of the probing result**: as the reviewer said, the probing result alone is not sufficient to determine factual association. We use probing results to qualitatively show that the behavior of model trained on referencing text is consistent with true factual associations and inconsistent with co-occurrence statistics, therefore the model likely learned true factual associations. We then proceeded to use reasoning evaluations to confirm that it is so in the section that follows.
>
> **Log likelihood ratio**: we thank the reviewer for pointing out the confusion, as the lines before Figure 2 only defines the likelihood ratio, missing an explanation of the log likelihood ratio used in the graph. Log likelihood ratio simply means the comparison ratio and negation ratio in log scale. We will clarify this in the revision of the paper.
>
> We will also appreciate any further feedback or discussion from the reviewer, and are glad to provide additional information and clarification.

---

> > ### Comment · Reviewer_hFpF · 2024-08-10
> >
> > Thanks the authors for their response. Since my score is already positive, I will keep my score.

---

### Official Review · Reviewer_iMfh · 2024-07-13

**Soundness:** 3
**Presentation:** 3
**Contribution:** 3
**Rating:** 5
**Confidence:** 4

**Summary:**

The work investigates the deficiencies of pretrained language models in learning factual knowledge, highlighting that these models tend to learn word co-occurrence statistics rather than true factual associations. The authors find that language models, when dealing with explicit relationships, are prone to merely memorize word co-occurrences and perform poorly on tasks that require reasoning.

**Strengths:**

* This work shows that language models tend to learn word co-occurrence statistics instead of true factual associations. This finding is important for improving the knowledge learning of language models.
* The authors propose two methods to improve the learning of factual associations. First, by using text with implicit rather than explicit factual associations, they force the model to learn these associations. Second, by actively forgetting the learned co-occurrence statistics, they allow the model to better learn and retain factual associations.
* The proposed strategies significantly improve the model's performance in multi-hop reasoning tasks on both synthetic and real-world datasets, proving their effectiveness.

**Weaknesses:**

* The generalization across different domains. This work synthesizes Country-City-Animal data, which is somewhat limited.
* Reasoning or memory? The purpose of implicit training is to force the model to understand factual associations through indirect connections, thereby enhancing its reasoning abilities. This approach will help the model perform better on complex, multi-step reasoning questions rather than simple memory tasks because of their training pattern. While, it can’t directly prove that referencing method can bring better memory than Co-occurrence. Moreover, for simple QA tasks, the Referencing method performs worse than the Narrative method. Different test tasks should be designed to verify knowledge retention. For instance, adding more noise and interference during simple QA tests to evaluate the robustness of memory. Design memory retrieval tasks that do not require complex reasoning to ensure that the tests only assess the model's ability to recall facts.
* Although it mentions that co-occurrence statistics and factual associations are parameterized in different layers of the Transformer model, it lacks a deep explanation of the specific mechanisms and reasons behind these phenomena.

**Questions:**

See weaknesses

**Limitations:**

Yes

---

> ### Author Rebuttal · Authors · 2024-08-07
>
> We thank the reviewer for the insightful comments, and we really appreciate the suggestions. Please find our response to the comments below:
>
> **Generalization of results across domains**: although we performed analysis mainly with the synthetic Country-City-Animals dataset (in order to ensure the knowledge is completely novel to the model, which helps us to cleanly ablate and evaluate learned knowledge in analysis), we also experimented on real-world datasets such as MQuAKE and 2WikiMultiHopQA in Section 4 which validated the effectiveness of our method.
>
> **Reasoning or memory**: we agree that memorization and reasoning (generalization) are two different aspects of knowledge learning. We thank the reviewer for raising this point and we will more clearly discuss the connection and difference between the two aspects in the revision of the paper.
> In existing work on knowledge editing, there have been multiple metrics designed to evaluate memorization, i.e., the model's ability to recall edited facts, such as varying the context and/or introducing semantic variations, similar to what the reviewer suggested. In our work, we mainly focus on the distinction between **memorization vs. reasoning** (generalization) and show that learned knowledge does not automatically generalize well to reasoning despite good memorization and the reasons behind it. We believe our work is a step forward from existing work on knowledge memorization and provides a new perspective on knowledge learning.
>
> **Explanation of the parameterization phenomenon**: we agree that investigating the root cause of the different parameterization would help better understand the LM's behavior in knowledge learning. Model inspection and interpretability methods could be leveraged in this direction. We would leave this as future work and will discuss this in the limitation section of the paper.
>
> We will also appreciate any further feedback or discussion from the reviewer, and are glad to provide additional information and clarification.

---

> > ### Comment · Reviewer_iMfh · 2024-08-11
> > **Response to Authors**
> >
> > Thanks for your rebuttal and it addresses most of my question. I will keep the same score.

---

### Official Review · Reviewer_B3nU · 2024-07-21

**Soundness:** 3
**Presentation:** 4
**Contribution:** 3
**Rating:** 7
**Confidence:** 4

**Summary:**

This paper investigates the learning of factual knowledge in pretrained language models, distinguishing between knowledge represented as word co-occurrence statistics and true factual associations. The authors find that language models tend to learn co-occurrence statistics, which do not generalize well to reasoning tasks, while factual associations, which generalize better, can be harder to learn. They propose two strategies to improve the learning of factual associations: training on text with implicit associations and using a method called active forgetting to discard learned co-occurrence statistics. Their experiments on synthetic and real-world datasets demonstrate that these strategies significantly enhance the models' ability to generalize factual knowledge in various reasoning scenarios. The paper includes a thorough layer-wise analysis of knowledge parameterization in transformer models finding different localization for co-occurence statistics vs factual knowledge in model weights.

**Strengths:**

I think the strengths of this paper are in the following contribtions

- Identification of Knowledge Representations: The paper clearly distinguishes between two forms of knowledge representation in language models: co-occurrence statistics and true factual associations. This distinction is crucial for understanding the limitations of current models. Additionally, the detailed analysis of how co-occurrence statistics and factual associations are parameterized across different layers of transformer models provides valuable insights into the internal workings of pretrained models.

- Empirical Validation: The authors conduct comprehensive experiments using synthetic and real-world datasets to validate their claims. They show that models trained on implicit associations generalize better to reasoning tasks than those trained on explicit co-occurrence.

- Novel Training Strategies: They propose a training strategies to improve factual learning are innovative. Training on text with implicit associations and a method of actively forgetting learned co-occurrence statistics to unblock factual learning.

- Public Release of Resources: Finally, the release of the synthetic corpus and code to reproduce their reulsts can facilitate further research and experimentation in this domain.

**Weaknesses:**

I did not find any major weaknesses in this paper.

The main ones, which are mentioned by the authors when addressing current limitations of their work are the following:

- Synthetic data split: how are you splitting your synthetic data? Are you evaluating on an unseen subset for both synthetic as well as natural dataset? I understood you are testing on unseen data for natural dataset and I am unsure if that's also the case for the synthetic dataset. Please clarify. This is the reason why I am, at the moment, giving a score of 6 for what would otherwise be a clear 7.

- Overhead in Data Preparation: Converting general text to forms with implicit associations for real-word data may require significant effort and sophisticated rewriting techniques, potentially limiting practical applicability.

- Limited Scope of Text Variations: The paper only considers two forms of text (narrative and implicit association). There is a need to explore more diverse textual representations to validate the findings comprehensively.

- Focus on a single type of reasoning: While the claims that learning implicit knowledge improve performance on complex reasoning tasks, the paper focuses on a specific type of reasoning. Other type of reasoning like logical or mathematical should be validated. Additionally, it is unclear whether the proposed finetuning method and data harm existing model performance on standard LLM benchmark. It would a nice addition to show whether the method in the paper do not conflict with existing model knowledge in other domains.

- Evaluation information: Taken from the appendix "For evaluation on question answering tasks, we report 5-shot exact match accuracy unless otherwise specified." Please add this in the main body of the paper and mention why you use this metric instead of others like F1 for QA tasks. Is it because all your tasks require a single word as gold label? Is this true also for the real-world dataset in table 3 (MQuAKE-T and 2WikiMultiHopQA)? Please add this info together with your generation parameters used at inference time (number of generated tokens/sampling parameters etc.)

-
---

Minor

- Missing reference: De Cao et al. Editing Factual Knowledge in Language Models, EMNLP 2021. This is an important reference when discussing model editing since it was among the first contribution in this area.

- line 200 the reference to Appendix 3.3 is wrong

----

### Final Recommendation

Overall, I think the claims are backed by well-presented empirical evidence and I vote for the inclusion of this paper to NeurIPS.

### Update post rebuttal

I increase my score from 6 to 7

**Questions:**

- Have you tried evaluating the model in a 0-shot fashion? Given the model has been finetuned on that data it can be helpful to add 0-shot performance

- How do you compute shaded areas in figure 3? For instance, it seems that MC accuracy of Llama 3 70B Narrative trained does not show decrease performance on the lowest layer for first-to-last ablation while it does for last-to-first ablation, yet you shaded that area for both ablation. It can be informative to add additional info on the criteria you used to shade those areas

- To compute the comparison ratio, the score depends on the choice of the entity in the denominator. Given the small size of your synthetic data, unless you are already doing so,, can you marginalize across all other entities? Please clarify how you compute the comparison ration

**Limitations:**

Yes in the limitations section after the conclusion on page 9

---

> ### Author Rebuttal · Authors · 2024-08-07
>
> We thank the reviewer for the insightful comments, and we really appreciate the suggestions. Please find our response to the comments below:
>
> **Data split**: The training and evaluation data are always disjoint and are of different types. The training data is plain text, while the evaluation data is questions answering of different reasoning types (we described the synthetic dataset in detail in Appendix A.1). We will try to make this more explicit in the revision of the paper.
>
> **Overhead in Data Preparation**: we agree that converting general text to forms with implicit associations may require significant effort. However, we think it is possible that LLMs such as ChatGPT could be prompted to perform such conversions automatically. We will try to investigate automatic methods for text rewriting in future work.
>
> **Scope of Text Variations**: we agree that the text variations are limited in the current work. Finding other forms of knowledge-bearing text would help generalize the findings.
>
> **Types of reasoning**: we included several types of reasoning in the evaluation, including reverse reasoning, indirect reasoning, and 2-hop reasoning. Indirect reasoning requires basic logical comparison between properties of two entities. Regarding mathematical reasoning, it would require numerical facts which are not included in the current dataset. We agree that it would be an interesting extension and we will consider in future work.
>
> **Conflict with existing model knowledge**: we use the MMLU benchmark to evaluate the existing model knowledge in other domains, and found that on the synthetic dataset, training on the narrative text results in 5-shot MMLU accuracy 66.7 -> 66.5 (Llama 3 8B) 79.2 -> 79.3 (Llama 3 70B), and training on text with implicit associations results in 66.7 -> 66.7 (Llama 3 8B) 79.2 -> 79.0 (Llama 3 70B). The performance change on MMLU is minimal and seems within margin of error in evaluation, indicating that either training method has little direct interference on the existing model knowledge (catastrophic forgetting is not observed most likely because the training data size is small).
> We will include evaluation on more common benchmarks in the revision of the paper.
>
> **Evaluation details**: we use exact match as metric for QA mainly because all answers in the datasets used are either single words (synthetic, 2WikiMultiHopQA) or entity names (synthetic, MQuAKE, 2WikiMultiHopQA). Generation uses greedy decoding and stops whenever '\n' is generated ('\n' follows the answer of each question in the 5-shot context) or a maximum of 20 tokens is reached. We will include these details in the revision of the paper and move the details about the metrics to the main text for better clarity.
>
> **Missing reference**: we thank the reviewer for pointing out the missing reference. We will include the reference in the revision of the paper.
>
> **0-shot evaluation**: the models are base LMs finetuned only on text corpora (narrative or referencing), and are not instruction-finetuned or finetuned on any QA data. Therefore, we don't expect the models to perform well on 0-shot evaluation on QA tasks, which would not effectively evaluate the model's knowledge or reasoning.
>
> **Criteria for shaded areas in Figure 3**: the shaded area shows the layers of significant performance change averaged over the two ablation directions. The shades are marked qualitatively and is mainly intended as a visual aid to help interpret the ablation curves.
>
> **Computation of the comparison ratio**: the negative samples (entity in the denominator) are chosen from entities of the same type from the dataset, excluding the correct entity itself. The log likelihood ratio is averaged over all possible choices.
>
> We will also appreciate any further feedback or discussion from the reviewer, and are glad to provide additional information and clarification.

---

> > ### Comment · Reviewer_B3nU · 2024-08-08
> > **Response to rebuttal**
> >
> > I want to thank the authors for their work on the rebuttal.
> >
> > I am glad to see that their method does not impact model existing knowledge as shown by the MMLU results.
> >
> > The rebuttal addressed most of my concerns. I still believe other type of reasoning tasks and different text variations would make the paper much stronger and are also important contribution for future work. However, the paper as is provides a valid contribution on understanding learning dynamics in LLMs. I am happy to increase my score to 7 and I vote for the inclusion of this work in the conference

---

### Decision · Program_Chairs · 2024-09-25

**Decision:**

Accept (spotlight)

**Comment:**

The paper presents research on how language models learn factual knowledge, emphasizing the distinction between word co-occurrence statistics and true factual associations, and proposes novel methods to improve the learning of factual associations in language models.  The paper conducts comprehensive experiments with both synthetic and real-world datasets.  The paper is written and presentation with clear structure, well-crafted approach and significant empirical findings.  The authors well addressed the concerns raised by the reviewers in the rebuttals.